# Tulathromycin and Diclazuril Lack Efficacy against *Theileria haneyi*, but Tulathromycin Is Not Associated with Adverse Clinical Effects in Six Treated Adult Horses

**DOI:** 10.3390/pathogens12030453

**Published:** 2023-03-14

**Authors:** Cynthia K. Onzere, Morgan Hulbert, Kelly P. Sears, Laura B. A. Williams, Lindsay M. Fry

**Affiliations:** 1Department of Veterinary Microbiology and Pathology, Washington State University, Pullman, WA 99164, USA; 2College of Veterinary Medicine, Cornell University, Ithaca, NY 14853, USA; 3Department of Clinical Science, Carlson College of Veterinary Medicine, Oregon State University, Corvallis, OR 97331, USA; 4Animal Disease Research Unit, USDA-ARS, Pullman, WA 99164, USA

**Keywords:** *Theileria haneyi*, *Theileria equi*, tulathromycin, safety, diclazuril, equine, adult

## Abstract

Equine theileriosis, caused by *Theileria haneyi* and *Theileria equi*, leads to anemia, exercise intolerance, and occasionally, death. Theileriosis-free countries prohibit the importation of infected horses, resulting in significant costs for the equine industry. Imidocarb dipropionate is the only treatment for *T. equi* in the United States, but lacks efficacy against *T. haneyi*. The goal of this study was to assess the in vivo efficacy of tulathromycin and diclazuril against *T. haneyi*. Fourteen *T. haneyi*-infected horses were utilized. Six were treated with eight weekly 2.5 mg/kg doses of tulathromycin. Three were treated daily for eight weeks with 2.5 mg/kg diclazuril. Three were pre-treated with 0.5 mg/kg diclazuril daily for one month to determine whether low-dose diclazuril prevents infection. Following infection, the dose was increased to 2.5 mg/kg for eight weeks. Two infected horses remained untreated as controls. The horses were assessed via nested PCR, physical exams, complete blood counts, serum chemistry panels, and cytology. Tulathromycin and diclazuril failed to clear *T. haneyi* and the treated and control groups exhibited similar parasitemia and packed cell volume declines. To obtain additional safety data on tulathromycin use in adult horses, necropsy and histopathology were performed on tulathromycin-treated horses. No significant lesions were detected.

## 1. Introduction

Equine theileriosis is caused by the tick-borne hemoparasites *Theileria haneyi* and *Theileria equi* [1,2,3,4]. *Theileria* sp. belongs to the phylum Apicomplexa, as do *Sarcocystis neurona*, the causative agent of equine protozoal myeloencephalitis (EPM), and *Plasmodium* sp., some of which are causative agents of malaria [5]. Equine theileriosis causes varying degrees of anemia, leading to lethargy, exercise intolerance, and potentially death in severe cases [2,3,6,7,8]. Studies indicate that *T. haneyi* infections lead to milder clinical disease than *T. equi*, such that most splenectomized horses infected with *T. haneyi* survive, whereas *T. equi* infection of splenectomized horses is almost invariably fatal [9,10]. *T. equi* infection precludes the transport, sale, import and export of horses into *T. equi*-free countries such as the U.S., leading to significant economic costs to the equine industry [3,10]. While the U.S. is currently free of *T. equi*, *T. haneyi* has been detected in horses detained at the southern U.S. border, and studies are currently underway to determine whether *T. haneyi* is present within U.S. horse populations. Co-infection of horses with both *T. equi* and *T. haneyi* has been documented in field samples from horses detained at the southern U.S. border and in controlled experimental settings [11]. 

Presently, imidocarb dipropionate (ID) is the only FDA-approved therapy for the clearance of *T. equi* in the U.S. and has consistently demonstrated efficacy against strains implicated in previous U.S. outbreaks of *T. equi* [12,13,14]. Unfortunately, ID is ineffective against *T. haneyi*, for which there are no currently known effective chemotherapeutics [11]. Additionally, in a significant proportion of horses co-infected with *T. equi* and *T. haneyi*, ID treatment failed to clear either parasite [11]. Thus, novel therapeutic strategies are urgently needed for equine theileriosis. The goal of the present study was to assess the in vivo efficacy of two commercially available, FDA-approved veterinary medications, tulathromycin (Draxxin^®^, Zoetis Inc., Kalamazoo, MI, USA) and diclazuril (Protazil, Merck Animal Health, Intervet Inc., Madison, NJ, USA), against *T. haneyi*.

Tulathromycin is a long-acting macrolide antimicrobial drug that inhibits bacterial protein synthesis via 50S ribosomal subunit binding [15,16]. Because of its unique three-amine functional group structure, tulathromycin proved to be efficacious in mitigating *Plasmodium yoelli* parasitemia in mice [17]. Furthermore, tulathromycin inhibited the in vitro growth of *T. equi*, supporting its potential for in vivo anti-protozoal efficacy in larger mammals [18]. Presently, tulathromycin has been approved for the treatment of bovine and porcine respiratory disease caused by Gram-positive bacteria and a subset of Gram-negative bacteria, including *Mannheimia haemolytica*, *Pasteurella multocida*, and *Mycoplasma* spp. In foals, certain macrolides, including tulathromycin, have been utilized in an extra-label fashion for the treatment of *Rhodococcus equi* infection [19]. However, as horses mature and their gastrointestinal tract develops enhanced hind-gut fermentation capabilities, the colonic microbiome extensively changes. In adult horses, treatment with older-generation macrolide antimicrobials often leads to severe, occasionally fatal, colitis due to negative effects on the gastrointestinal flora [20,21,22]. 

Recently, Leventhal et al. studied the pharmacokinetics and safety of tulathromycin administration in a small group of adult horses [23]. In this study, none of the horses developed colitis; however, the administration of tulathromycin via subcutaneous or intramuscular administration led to substantial and mild discomfort, respectively. No adverse reactions were observed when the dose of tulathromycin was diluted to sixty milliliters of sterile saline and intravenously administered over fifteen minutes [23]. 

Diclazuril is a triazine anti-protozoal drug that is currently FDA-approved for the treatment of EPM due to *Sarcocystis neurona* or less commonly, *Neospora hughesi*, and is easily accessible to U.S. producers as Protazil (1.56% ponazuril pellets, Merck Animal Health, Madison, NJ, USA) [24,25,26]. When used at an extra-label daily dose of 0.5 mg/kg, diclazuril reached adequate plasma concentrations to prevent EPM [27,28]. In a previous in vitro study, the related drug, ponazuril, showed efficacy against *T. equi* at high concentrations [29]. While ponazuril is only available as a paste, diclazuril is available as a palatable food topper, and is thus easier to administer for extended periods. For this reason, and because *Sarcocystis* and *Theileria* are sister taxa, sharing a recent common ancestor [30], diclazuril is promising as a potential therapeutic agent for *T. haneyi*.

Herein, we present data on the efficacy of diclazuril and tulathromycin to clear horses persistently infected with *T. haneyi.* We also present data regarding the safety profile of tulathromycin treatment in adult horses. 

## 2. Materials and Methods

### 2.1. Horses

Eight adult (1–16 years old) Welsh pony crosses (7 mares and 1 stallion) and six adult (5–15 years old) mixed-breed horses (two mares and four geldings) in research herds maintained at the University of Idaho or Washington State University were utilized in this study. The horses were divided into four experimental groups, as shown in Table 1 below. Prior to the study, horses received routine immunizations for respiratory viruses, tetanus, encephalitis viruses and West Nile virus, were dewormed with label doses of fenbendazole, and were treated for ectoparasites with label doses of Clean Up 3% diflubenzuron and 5% permethrin (Elanco Animal Health, Indianapolis, IN, USA). Approximately 1.5 months before the onset of the study, several of the horses contracted equine influenza, which was resolved with supportive care. Prior to inoculation, the horses were deemed healthy via physical examination by a licensed veterinarian and evaluation of complete blood count and serum chemistry panel. Horses were intravenously inoculated with a single cryovial (~1.2–1.8 mL) of *T. haneyi* (eagle pass isolate)-infected erythrocyte stabilate derived from the blood of an experimentally infected horse [1], as shown in Table 1. Larger horses in Group 1 were inoculated with a stabilate of higher percent parasitized erythrocytes than the small ponies that comprised Groups 2, 3, and the control group. Prior to inoculation, the erythrocyte stabilate was thawed slowly at room temperature, combined with autologous serum, and then intravenously injected over several minutes. All the horses were monitored for 30 min following infection for anaphylaxis, which was not observed. For all the horses, successful *T. haneyi* infection was confirmed using nested PCR (nPCR, see below). These animal experiments were approved by the Washington State University and University of Idaho Institutional Animal Care and Use Committees, ASAF number 6982 (WSU) and 2021-43 (UI). 

### 2.2. Post-Infection Monitoring and Assessment of Parasitemia

Following *T. haneyi* infection, rectal temperature, heart rate, respiratory rate, and overall attitude and appetite of each horse were assessed daily. Blood was collected at successive timepoints throughout the experiment via jugular venipuncture to assess the packed cell volume (PCV) and percent parasitized erythrocytes (PPE), and for complete blood count (CBC), serum chemistry panels, and *T. haneyi* nPCR. For these analyses, blood was collected into ethylenediaminetetraacetic acid (EDTA) and serum separator vacutainer tubes. Once the blood had clotted, serum separator tubes were centrifuged at 1800× *g* for 10 min, and serum was removed from the clot for analysis. PPE was determined via the evaluation of Giemsa-stained blood smears using the following equation: ((Total parasites in 5 fields)/(Erythrocyte count in ¼ of a field × 20)) × 100.

### 2.3. T. haneyi Nested PCR

Genomic DNA was isolated from whole blood using the DNeasy Blood and Tissue Kit (Qiagen, Inc., Hilden, Germany) in accordance with the manufacturer’s instructions. A negative extraction control that contained 1X PBS as a sample was used to assess contamination during DNA extraction. Nested PCR that targeted a unique gene for the detection of *T. haneyi* was performed as previously described [9]. Briefly, the primary PCR was performed using 12.5 µL of the DreamTaq Green PCR Master Mix (2X) (Thermo Scientific™, Waltham, MA, USA), 10µM each of the external forward (5′ CCATACAACCCACTAGAG 3′) and reverse (5′ CTGTCATTTGGGTTTGATAG 3′) primers, 5.5 µL of PCR-grade water and 5 µL of the extracted DNA at 20–100 ng/µL per sample. Genomic DNA from a previously infected horse confirmed to be *T. haneyi*-infected via both light microscopy and nPCR was used as a positive control, and a negative template control comprising of PCR-grade water was used to assess contamination during the PCR process. The primary PCR reaction was performed using previously described thermal cycling conditions [9]. In the secondary PCR reaction, 1 µL of the primary PCR product was used as a template, alongside 12.5 µL of the DreamTaq Green PCR Master Mix (2X), 10 µM each of the internal forward (5′ GACAACAGAGAGGTGATT 3′) and reverse (5′ CGTTGAATGTAATGGGAAC 3′) primers and 9.5 µL of PCR-grade water. The secondary PCR reaction was also performed using previously described thermal cycling conditions [9], resulting in the amplification of a 238 bp product and a 1% agarose gel was used for visualization of the PCR product. 

### 2.4. Tulathromycin Treatment

To allow horses to reach the persistent phase of infection, treatment was not initiated until one month after infection confirmation by nPCR. At this point, horses in Group 1 were treated weekly with 2.5 mg/kg tulathromycin (100 mg/mL, Draxxin^®^, Zoetis, Parsipanny, NJ, USA) diluted to 60 mL in sterile saline and administered intravenously via jugular venipuncture over 15 min. This dosage was chosen due to previous evidence of its safety and efficient tissue distribution in adult horses [23]. Each horse received 8 weekly treatments, with a four-week break between treatments 4 and 5 to minimize potential complications from long-term antibiotic treatment. Following each treatment, horses were closely monitored for 72 h for signs of diarrhea, reduced appetite, or depression, colic, or fever. If horses developed diarrhea or signs of colic, they were examined by a veterinarian and treated as deemed necessary. Mild colic was treated with 0.55–1.1 mg/kg of flunixin meglumine administered via IV (Prevail^TM^, Vet One, Boise, ID, USA). 

### 2.5. Diclazuril Treatment

One month after the confirmation of successful *T. haneyi* infection via nPCR (to allow horses to reach the persistent phase of infection), horses in Group 2 started daily treatment with 2.5 mg/kg 1.56% of diclazuril pelleted food topper (Protazil^®^, Merck Animal Health, Madison, NJ, USA) for eight weeks, with a four-week break between the fourth and fifth weeks of treatment. A dose of 2.5 mg/kg was selected due to evidence that the related drug, ponazuril, inhibited *T. equi* replication in vitro, but only at high drug concentrations. The label dosage for diclazuril is 1 mg/kg; however, the administration of 2.5 mg/kg did not elicit adverse reactions in the initial drug safety trials. 

Due to evidence that the pre-treatment of horses with low doses of diclazuril (0.5 mg/kg twice a week) maintains plasma levels sufficient to inhibit *S. neurona* [27,28], horses in Group 3 were treated daily for 30 days prior to infection with 0.5 mg/kg 1.56% of diclazuril pelleted food topper. Following infection, the dose was maintained at 0.5 mg/kg per day until 30 days after the horses became nPCR-positive for *T. haneyi*, at which point the dose was increased to 2.5 mg/kg per day for eight weeks, with a one-month break between weeks 4 and 5 of treatment.

### 2.6. Necropsy and Histopathology

Following euthanasia, complete necropsy was performed by two boarded Veterinary Anatomic Pathologists on all horses in group 1. During necropsy, the tissues grossly evaluated and sampled for histopathology included the following: brain, pituitary gland, tongue, trachea, esophagus, thyroid gland, heart, lungs, diaphragm, liver, spleen, kidneys, adrenal glands, stomach, duodenum, jejunum, ileum, cecum, and colon. Tissues grossly examined but not sampled for histopathology included the integument, eyes, teeth, guttural pouches, reproductive tract (mares), urinary bladder, and bone marrow. In some cases, additional gross lesions were detected and sampled for histopathology (see Section 3). Tissues were fixed in 10% neutral buffered formalin for approximately 30 days, and then formalin-fixed specimens were trimmed, routinely processed, stained with hematoxylin and eosin (H&E), and examined independently by both boarded Veterinary Anatomic pathologists using light microscopy. 

## 3. Results

### 3.1. Theileria haneyi Infection Outcome

Following intravenous inoculation with *T. haneyi* stabilate, all horses except one (Ho-124) were nested PCR-positive for *T. haneyi* by day 14 (Ho-124 was nPCR-positive on day 20 post-infection). In all horses, percent parasitized erythrocytes remained below 0.2% for the entire experiment, with only the rare observation of merozoite-infected erythrocytes (Figure 1), consistent with previous observations of *T. haneyi* in spleen-intact horses. Horses exhibited a mild decline in packed cell volume (PCV) over time (Figure 2); however, only four horses (431, 420, 419, and 124) developed anemia (defined as a PCV < 30%), with PCV nadirs of 29%, 26%, 23%, and 29%, respectively. None of the horses exhibited pyrexia. Changes in the leukogram parameters, including monocytosis and lymphocytosis, were intermittent and consistent with those noted in previous studies. 

### 3.2. Tulathromycin Treatment Outcome

Beginning 30 days after they became nPCR-positive for *T. haneyi*, horses in Group 1 were treated weekly for eight weeks with 2.5 mg/kg tulathromycin administered intravenously. To minimize the potential adverse effects of long-term antibiotic administration, horses were rested for four weeks between the fourth and fifth treatments. All horses in Group 1 remained positive for *T. haneyi* throughout the experiment (Table 2). Horses in both Group 1 and the control group exhibited similar, mild declines in mean PCV, with nadirs of 34% (±1.5) and 33% (±2.8), respectively (Figure 2). Rare organisms were detected in blood smears of individual animals in each group throughout the experiment (Figure 1). 

### 3.3. Tulathromycin Safety

Due to the risk of severe colitis and injection site reactions caused by macrolide antibiotic administration in adult horses [20], horses in Group 1 were closely monitored following each tulathromycin treatment for signs of injection site reactions (e.g., redness, pain, and swelling), diarrhea, colic (e.g., kicking abdomen, flank watching, sweating, pacing, rolling and anorexia), and systemic inflammation/sepsis (pyrexia, tachycardia, tachypnea, depression, mucous membrane injection, presence of a toxic line, leukopenia, neutropenia, leukocytosis and neutrophilia). None of the horses developed injection site reactions (Table 3). Two horses exhibited mild to moderate, transient clinical signs consistent with colic during the experiment (Table 3). Horse 427 had the following two mild clinical issues: on the day of the third dose, this horse had a brief episode of sweating followed by soft fecal consistency for less than 12 h. Her appetite remained normal, and the episode was resolved without intervention. On the day of the fifth dose, within one hour of tulathromycin treatment, the horse exhibited clinical signs consistent with mild abdominal discomfort that was resolved with a single dose of flunixin meglumine. During this episode, her appetite, temperature, pulse, and respiratory rate were normal. 

Horse 424 had three episodes of clinical discomfort; two were of mild severity and one moderate. Four days after the third tulathromycin treatment, this horse developed mildly soft fecal consistency for two days that was resolved without intervention. There was no evidence of abdominal pain, nor was there elevation in rectal temperature, pulse, or respiratory rate during this episode. The second episode was observed two hours after the fourth tulathromycin treatment, when the horse exhibited mild abdominal discomfort that was resolved with a single dose of flunixin meglumine. During this episode, his appetite, temperature, pulse, and respiratory rate were normal. The third episode lasted approximately one week. The day of the fifth treatment, the horse developed mild diarrhea (cow-pie consistency) that lasted for three days before the fecal consistency returned to normal. During this time period, the horse had a normal attitude and appetite, as well as a normal temperature, pulse and respiratory rate. No intervention was required to maintain hydration. Five days after the fifth treatment, the horse developed signs of moderate abdominal discomfort that warranted closer monitoring, and he was admitted to the Washington State University Veterinary Teaching Hospital Equine Medicine Service out of caution due to his previous GI signs. Upon admission, his temperature, pulse, respiratory rate, PCV, capillary refill time, mucous membrane color, CBC, chemistry panel and lactate levels were normal, and he was subsequently diagnosed with a soft pelvic flexure impaction via ultrasound and rectal palpation. The horse was managed with enteral fluid therapy, exhibited no further signs of colic, and passed loose stool that gradually regained normal consistency over the next 24 h. 

In addition to these clinical issues, horses in Group 1 also exhibited several intercurrent (unrelated) clinical issues throughout the course of the experiment (Table 4), including intermittent urticaria that was partially improved with fly spray application, a cutaneous foreign body (stick) that was resolved with stick removal and wound flushing, cutaneous papillomas on the muzzle, soft tissue mass on the chest, transient cough and nasal discharge, a skin wound with granulation tissue and intermittent swelling, and intermittent ventral edema (“stocking up”) that was resolved with exercise.

To better characterize the safety profile of tulathromycin in adult horses, regular CBC and serum chemistry panel assessment was completed throughout the experiment. The mean white blood cell counts for Group 1 and for the control group were within normal limits throughout the experiment, and mean values for the groups were similar throughout the experiment (Appendix A). Similarly, mean levels of gamma glutamyl transferase (GGT, Appendix A), total bilirubin (Appendix A), albumin (Appendix A), blood urea nitrogen (BUN, Appendix A), and creatinine (Appendix A) were within normal limits for both groups throughout the experiment. The mean serum globulin level for Group 1 was consistently higher than that of the control group throughout the experiment (Appendix A), with mean levels in the elevated range (>4.7 g/dL) at multiple timepoints throughout the experiment (peak mean level 4.8 ± 0.52 g/dL). Due to the elevated mean globulin levels, the mean albumin: globulin ratio for Group 1 was often lower than that of the control group throughout the experiment, and was <0.7 at most timepoints (Appendix A). 

To further support the safety of tulathromycin for adult horses, complete necropsy and histopathology were performed on all horses in Group 1 following euthanasia. Importantly, necropsy and histopathology revealed marked muscular hypertrophy of the ileum in horse 424, along with low numbers of embedded cyathostome parasites. All horses exhibited mild lymphoplasmacytic and eosinophilic infiltration of the lamina propria judged to be within normal limits for adult horses [31,32]. In few foci, eosinophils were subjectively increased, likely to be secondary to mild gastrointestinal parasitism. All other findings were considered to be incidental (Table 5). 

### 3.4. Diclazuril Treatment Outcome

Beginning 30 days after they became nPCR-positive for *T. haneyi*, horses in Group 2 were treated for four weeks with 2.5 mg/kg diclazuril administered orally (PO). Horses were then rested for four weeks and subsequently treated with the same dose for an additional four weeks. Treatment length was selected based on the four-week treatment course necessary to clear *Sarcocystis neurona.* All horses in Group 2 remained positive for *T. haneyi* throughout the experiment (Table 6). Rare organisms were detected on blood smears throughout the experiment in horses of both Group 2 and the control group (Figure 1), and horses in both groups exhibited a similar, mild decrease in mean PCV, with nadirs of 29% (±2.3) and 33% (±2.8), respectively (Figure 2). 

Based on previous studies showing that the pre-treatment of horses with diclazuril can prevent *S. neurona* [27,28], horses in Group 3 were pre-treated with a low dose (0.5 mg/kg PO) of diclazuril daily for 30 days prior to *T. haneyi* inoculation in order to determine whether low-dose diclazuril would prevent or delay *T. haneyi* infection. Following inoculation, 2/3 horses in this group were nPCR-positive for *T. haneyi* by day 14, and 1horse became nPCR-positive on day 20 (Table 7), suggesting that diclazuril pre-treatment is not efficacious in preventing or delaying *T. haneyi* infection. Once nPCR-positive, horses in this group were treated for another 30 days with daily doses of diclazuril at 0.5 mg/kg PO, and then the dose was increased to 2.5 mg/kg PO per day for eight weeks, with a four-week break between the fourth and fifth weeks of high-dose treatment. Treatment was paused between four-week, high-dose cycles to allow washout, minimize the risk of complications, and to allow the complete assessment of data regarding the outcome of the first treatment course. Horses in Group 3 remained nPCR-positive for *T. haneyi* for the remainder of the experiment (Table 7). Rare organisms were occasionally detected on blood smears in Group 3 and in the control group (Figure 1), and both groups exhibited a similar, mild mean decline in PCV, with nadirs of 31% (±7) and 33% (±2.8), respectively (Figure 2).

## 4. Discussion

These data indicate that tulathromycin and diclazuril are ineffective in vivo chemotherapeutic agents against *T. haneyi* at the dosages utilized in this study. Indeed, disease course and severity in the treated and control groups were similar, and all horses remained positive for *T. haneyi* via nPCR throughout the study. Previous in vitro data suggested that ponazuril, a drug closely related to diclazuril, could eliminate *T. equi*, but only at very high drug concentrations [29]. Thus, although diclazuril may simply lack efficacy against *T. haneyi*, another possible explanation for the results obtained with diclazuril in the present study is that the dose utilized was not sufficient to reach theileriacidal levels. The label dose recommended to eliminate the related parasite, *Sarcocystis neurona*, is 1 mg/kg PO for 28 days. Given the in vitro findings with ponazuril and *T. equi*, we elected to utilize diclazuril at 2.5 times this dose for 8 weeks, a high dosage still within the known margin of safety for diclazuril in horses. Since there are limited safety data that suggest 5 mg/kg PO for 42 days may not result in adverse effects, this experiment could be repeated using a higher dose of diclazuril to rule out potential anti-*T. haneyi* efficacy more definitively.

The dosage and route of tulathromycin utilized in this study were selected based on recent data demonstrating the sustained presence of the drug within plasma and pulmonary epithelial lining fluid without adverse effects when administered at 2.5 mg/kg IV to healthy adult horses [23]. These data suggest that 2.5 mg/kg, administered slowly via the intravenous route, is sufficient to achieve widespread and lasting drug distribution within the plasma and respiratory tract [23]. Given the sensitivity of the equine gastrointestinal tract to macrolide antibiotics in general [20], experimentation with this drug as a potential theileriacidal agent at higher dosages should be performed cautiously, with careful consideration of the risk vs. benefit analysis, given the low risk of morbidity and mortality associated with *T. haneyi* relative to that of colitis following aggressive macrolide antibiotic administration. Similarly, although subcutaneous and intramuscular administration could improve absorption and distribution, these routes of administration result in severe injection site reactions in horses [23], and should thus be avoided if possible.

In addition to the assessment of the in vivo efficacy of these drugs against *T. haneyi*, the other objective of this study was to expand the available dataset on the safety of tulathromycin in adult horses. Horses in Group 1 were treated eight times with intravenous tulathromycin. Despite the long treatment period and repeated administration of tulathromycin during that treatment period, none of the horses developed colitis. Two horses exhibited episodes of colic; however, in one horse, the episodes were very short and mild, and were resolved with no to limited intervention. The other horse had more repetitive bouts of colic and diarrhea, one of which was diagnosed as a pelvic flexure impaction and resulted in brief hospitalization and enteral fluid therapy. Interestingly, this horse was found to have severe muscular hypertrophy of the ileum via necropsy and histopathology. Ileal muscular hypertrophy is a rare condition associated with intermittent colic in horses [33,34]. The cause is unknown; occasionally affected horses have intestinal parasites, but causation has not been proven [33,34]. In this case, the horse did have a low level of encysted cyathostome parasites within his intestinal tract. It is likely that the repeated episodes of colic and diarrhea in this horse were caused or exacerbated by the muscular hypertrophy of his ileum. 

Apart from ileal muscular hypertrophy in horse 424, necropsy and histopathology revealed mild infiltration of the lamina propria of the small and large intestine by lymphocytes, plasma cells, and eosinophils at a level considered as normal for adult horses [31,32]. In a few areas, eosinophils were subjectively increased, which is common in horses with gastrointestinal parasitism (horses in this group had mild gastric myiasis and cyathostome infections). In addition, several incidental lesions were detected in some of the horses. These included mild lymphoplasmacytic periportal hepatitis and interstitial nephritis, and chronic, focal, mild inflammation or fibrosis within the lungs. These chronic, mild inflammatory lesions are commonly detected on postmortem examination of tissues from adult horses, and likely result from previous infections or inflammation, or from chronic, low-level antigenic stimulation. One horse had a microscopic pituitary cyst, one had cutaneous papillomas of the muzzle, and one had a sarcoma/sarcoid. Finally, two horses had low-level cyathostome infections, and two horses had gastric myiasis. All of these are common, incidental, findings in adult horses. There was no evidence of hepatic, renal, cardiac, pulmonary, integumentary, neurologic, or gastrointestinal lesions resulting from tulathromycin administration.

Overall, horses in Group 1 did not exhibit abnormal values in the analytes of complete blood counts or chemistry panel results, except for the mean serum globulin levels, which were higher than those of the control group at all timepoints. Elevated globulin levels led to decreased mean serum albumin: globulin ratios in Group 1 compared to the control group. This finding is likely related to the intercurrent clinical problems present in several horses in Group 1. For instance, horse 431 had a chronic laceration on his right rear leg that exhibited waxing and waning inflammation over the course of the experiment, horse 425 had an abscess caused by a cutaneous foreign body, and horse 427 developed transient upper respiratory infection. Additionally, several of the horses recovered from equine influenza one month prior to the onset of the experiment. No other lesions were detected at necropsy to explain the elevated globulin levels in these horses.

In conclusion, additional studies to identify safe and efficacious anti-theilerial drugs for use in horses are urgently needed. Large-scale compound library screening experiments have enabled the identification of potentially efficacious drugs in some apicomplexan systems, and similar studies are needed for *T. haneyi* [35,36,37,38,39,40]. Furthermore, while several compounds have demonstrated in vitro efficacy against *T. equi*, very few have been tested against *T. haneyi*, or against either parasite in vivo [5,40,41,42,43]. Continued in vivo trials with promising drug candidates are necessary for the sustained control of equine theileriosis across the globe. Finally, our data provide further support that repeated intravenous administration of tulathromycin to adult horses does not result in adverse clinical effects, and that tulathromycin may be a useful antimicrobial drug for the treatment of susceptible bacterial diseases of equids. Larger safety trials of tulathromycin in adult horses may be warranted to allow the expansion of its use in equine medicine.

## Figures and Tables

**Figure 1 pathogens-12-00453-f001:**
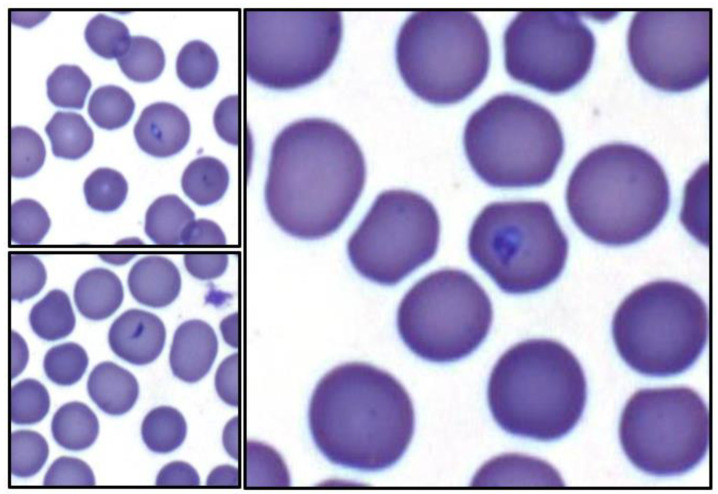
Representative merozoite-infected erythrocytes from *T. haneyi*-infected horses in Groups 1–3. Throughout the experiment, low levels (<0.2% parasitized erythrocytes) of parasitized erythrocytes were detected via evaluation of Giemsa-stained blood smears. Organisms were pear-shaped to elliptical shapes, and 1–3 microns long with a basophilic apical organelle complex.

**Figure 2 pathogens-12-00453-f002:**
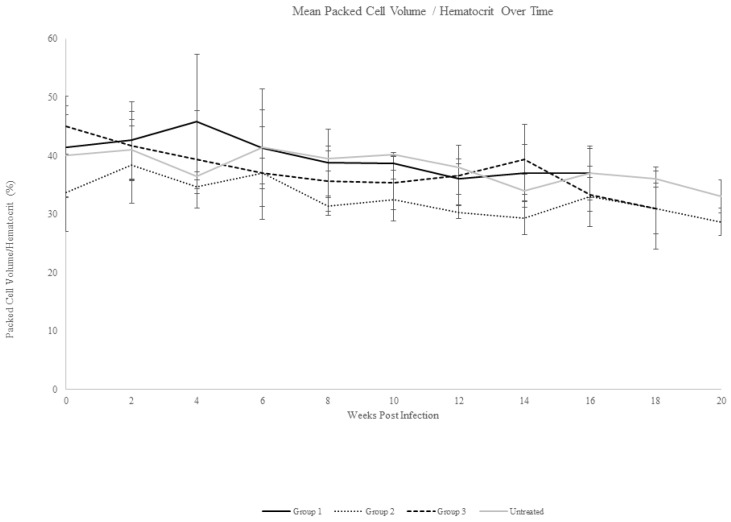
Mean packed cell volume/hematocrit over time. Mean erythrocyte levels for each experimental group, including the untreated control group, are shown. Mean change in PCV/HCT was similar across groups, and the PCV/HCT decline was mild.

**Table 1 pathogens-12-00453-t001:** Horses utilized in the study and *T. haneyi* erythrocyte stabilate used for inoculation. PPE = percent parasitized erythrocytes.

Horse	Sex	Age (years)	Breed	Treatment	Group	Stabilate
424	Gelding	8	QH X	Tulathromycin	1	Horse 344, 12.69% PPE
425	Mare	5	Morgan X			
426	Gelding	5	QH X			
427	Mare	5	QH X			
429	Gelding	8	TB X			
431	Gelding	15	QH X			
417	Mare	1	WP X	Diclazuril Post-Infection	2	Horse 344, 6.1% PPE
418	Mare	1	WP X			
420	Stallion	1	WP X			
124	Mare	16	WP X	Diclazuril Pre/Post-Infection	3	
173	Mare	11	WP X			
419	Mare	1	WP x			
411	Mare	2	WP X	None	Control	
413	Mare	2	WP X			

**Table 2 pathogens-12-00453-t002:** *Theileria haneyi* nested PCR results for horses in Group 1 following each tulathromycin treatment (Tx). Treatment number indicated after Tx.

Horse	Group	Pre-Infection	Pre-Tx	Post-Tx 1	Post-Tx 2	Post-Tx 3	Post-Tx 4	Post-Tx 5	Post-Tx 6	Post-Tx 7	Post-Tx 8
424	1	-	+	+	+	+	+	+	+	+	+
425	1	-	+	+	+	+	+	+	+	+	+
426	1	-	+	+	+	+	+	+	+	+	+
427	1	-	+	+	+	+	+	+	+	+	+
429	1	-	+	+	+	+	+	+	+	+	+
431	1	-	+	+	+	+	+	+	+	+	+

**Table 3 pathogens-12-00453-t003:** Summary of clinical findings for horses in Group 1.

Horse	Group	Diarrhea	Colic	Fever	Anorexia	Depression	Injection Site Pain
424	1	Mild, transient	Mild to moderate	-	-	-	-
425	1	-	-	-	-	-	-
426	1	-	-	-	-	-	-
427	1	-	Mild, transient	-	-	-	-
429	1	-	-	-	-	-	-
431	1	-	-	-	-	-	-

**Table 4 pathogens-12-00453-t004:** Incidental, intercurrent health issues for horses in Group 1.

Horse	Group	Intercurrent Health Issues
424	1	None
425	1	Cutaneous foreign body, intermittent urticaria
426	1	Papillomas on muzzle
427	1	Skin mass on chest, transient cough, and nasal discharge
429	1	None
431	1	Intermittent urticaria, scar with granulation tissue/intermittent swelling on right rear leg, intermittent ventral edema (“stocking up”)

**Table 5 pathogens-12-00453-t005:** Necropsy and histopathology findings for horses in Group 1, NSF = no significant findings.

Horse	Skin	Brain and Pituitary Gland	Heart	Lungs, Trachea	GI Tract *	Liver	Kidneys	Spleen
424	NSF	Pituitary cyst	NSF	NSF	Muscular hypertrophy of ileum, few cyathostomes	Mild periportal infiltrate	NSF	NSF
425	NSF	NSF	NSF	NSF	NSF	Mild periportal infiltrate	Mild interstitial nephritis	NSF
426	Muzzle papilloma	NSF	NSF	NSF	Encysted cyathostomes	Mild periportal infiltrate	Mild interstitial nephritis	NSF
427	Sarcoma/ sarcoid	NSF	NSF	Focal histiocytic infiltrate	Gastric bots	Mild periportal infiltrate	NSF	NSF
429	NSF	NSF	NSF	Multifocal pleural fibrosis	Mesenteric lipoma	NSF	NSF	NSF
431	Scar, right hock	NSF	NSF	NSF	Gastric bots	Mild periportal infiltrate	Mild interstitial nephritis	NSF

* All horses exhibited mild lymphoplasmacytic and eosinophilic infiltrates within the lamina propria of the small intestine and colon, with few areas of increased eosinophilic infiltrate. This finding was considered within the normal limits for adult horses with mild intestinal parasitism.

**Table 6 pathogens-12-00453-t006:** *Theileria haneyi* nested PCR results for horses in Group 2 throughout the experiment, including two-week intervals during diclazuril treatment (Tx). DPI = days post infection.

Horse	Group	Pre-Infection	14 DPI, Pre-Tx	2 Weeks of Tx	4 Weeks of Tx	6 Weeks of Tx	8 Weeks of Tx
417	2	−	+	+	+	+	+
418	2	−	+	+	+	+	+
420	2	−	+	+	+	+	+

**Table 7 pathogens-12-00453-t007:** *Theileria haneyi* nested PCR results for horses in Group 3 throughout the experiment, including two-week intervals during diclazuril treatment (Tx). DPI = days post infection.

Horse	Group	Pre-Infection	14 DPI	21 DPI	2 Weeks of Tx at 2.5 mg/kg	4 Weeks of Tx at 2.5 mg/kg	6 Weeks of Tx at 2.5 mg/kg	8 Weeks of Tx at 2.5 mg/kg
124	3	−	−	+	+	+	+	+
173	3	−	+	+	+	+	+	+
419	3	−	+	+	+	+	+	+

## Data Availability

All data are provided in the manuscript and Appendix A.

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
