# Peer review of "Tulathromycin and Diclazuril Lack Efficacy against Theileria haneyi, but Tulathromycin Is Not Associated with Adverse Clinical Effects in Six Treated Adult Horses"

_pathogens, 2023, doi:10.3390/pathogens12030453_

Round 1

Reviewer 1 Report

General comments - Minor Concerns:

The study entitled “Tulathromycin and diclazuril lack in vivo efficacy against Theileria haneyi infection in adult horses but tulathoromycin treatment is not associated with adverse clinical effects” reports the evaluation of the in vivo efficacy of either tulathromycin and diclazuril to treat the infection of Theileria haneyi (T. haneyi) and also the in vivo efficacy of diclazuril to prevent the infection of T. haneyi. T. haneyi has been recently detected in horses from US southern border and considering that equine theilerosis leads to significant economics costs to equine industry, the evaluation of new therapeutic strategies assessed in this work adds more information to the currently known therapy for these agents. Moreover, the manuscript is clearly written and the information throughout is very well organized. However, there are some major comments and few minor comments that the authors should address before this manuscript can be accepted for publication.

Introduction:

-        The paragraph in lines 54-57 is redundant with the one in lines 90-93, regarding the aim of the study.

Materials and methods:

Regarding the experimental groups and the infection and treatment design,

-        Why did the authors used different breeds and ages for group 1 and groups 2, 3 and Controls? Was a randomized design of groups performed, and if not, could authors explained the group design criteria?

-        Why group 1 was infected with stabilate of higher percent parasitized erythrocytes? Please clarify

-        Did the authors perform any statistical calculation on the number of horses assigned for each treatment group and controls?

-        Why did treatments in groups 1 and 2 started at 30 days’ post confirmation of T. haneyi infection?

-        Was there any statistical analysis performed? If so, please include it in the manuscript.

-        Table 1: please define PPE

-        Line 119: what the authors mean by “serially”

Results:

-        Lines 224-228: It is hard to analyzed the results of group 1 compared to C group when the design of the groups is not clearly stated. Authors stated that there are “no significant differences” in PPE or in PVC when compared G1 to GC. However, as mentioned before there is no explanation of the statistical analysis used for these study, and the randomization of the horses for each group.

-        Table 2: In “Post –Tx XX”, the number reflects the number of the week? Please clarify.

-        Table 6 and Table 7: Columns of “ 2 weeks, 4 weeks, etc…” should be after the period of 30 days post-detection of T. haneyi?? Please clarify.

Discussion:

This reviewer suggests to mention the convalescence of equine influenza of horses in “M&M” rather than in “Discussion” section. Moreover, as 2 horses had gastric bots and another 2 had cyathostomes as detected by necropsy, could the authors clarify if these horses from research herds had a antiparasitic programme and a sanitary vaccination programme?

Reviewer 3 Report

The manuscript refers the lack of in vivo efficacy of tulathromycin in treatment of Theileria haneyi infection together with diclazuril in both treatment and prevention of this piroplasmosis in adult horses. The topic is of interest due the drug resistance of this parasite.

Why did the Authors test diclazuril instead of ponazuril (previously successfully tested against Theileria equi)? Please, add information and modify accordingly the sentence at lines 85-89. 

some other minor concerns are listed below

The title of paper is too long and the lack of adverse effects should be deleted. The number of tested horses is very small

Lines 72-80 - please remove this sentence from introduction section

lines 61, 87, 172, 357,  363, 428 - in vitro 

lines 54 62, 90, 92, 354, 381, 429 - in vivo

Round 2

Reviewer 1 Report

The manuscript has been improved with author´s comments